# Prevalence and associated factors of occupational injuries among garment and textile workers: Evidence from the Bangladesh Labour Force Survey 2016–17

Md. Tariqujjaman[1]*, Arifa Farzana Tanha[1], Md. Alamgir Hossain[2], Abul Hares[3], Md. Matiur Rahaman[4], Nadia Sultana[1], Fahmida Ferdous[5], Md. Mehedi Hasan[6,7], Md. Rashidul Azad[8]

1 Nutrition Research Division, icddr,b, Dhaka, Bangladesh, 2 Maternal and Child Health Division, icddr,b, Dhaka, Bangladesh, 3 Health Systems and Population Studies Division, icddr,b, Dhaka, Bangladesh, 4 Department of Social Policy and Social Security Studies, Bonn-Rhein-Sieg University of Applied Sciences, Bonn, Germany, 5 Bangladesh Bureau of Statistics, Dhaka, Bangladesh, 6 Poche Centre for Indigenous Health, The University of Queensland, Brisbane, Queensland, Australia, 7 ARC Centre of Excellence for Children and Families over the Life Course, The University of Queensland, Brisbane, Queensland, Australia  8 House of Research and Development, Dhaka, Bangladesh

* md.tariqujjaman@icddrb.org

## Abstract

Annually, numerous workers face job loss, injuries, and fatalities due to various occupational injuries (OIs). However, less is known regarding the burden of OIs and their associated factors in the textile and garment industries in Bangladesh. This study aimed to determine the prevalence of OI and the individual and job-related factors associated with OI among textile and garment workers in Bangladesh. We analyzed cross-sectional data of 13,738 workers collected during 2016–2017 from the nationally representative Bangladesh Labor Force Survey. We employed multiple Firth logistic regression models to explore the different levels of associated factors of OI. The overall prevalence of OI was 1.8%, with a higher prevalence in the textile industry (3.8%) compared to the garment industry (1.2%). Within the textile industry, jute manufacturing exhibited the highest prevalence (12.3%), while in the garment sector, the embroidery and wearing industries had the highest prevalence (1.8%). Adjusted models revealed that, in the textile industry, migrant workers had higher odds of OI (Adjusted Odds Ratio, AOR = 1.65; p = 0.017) compared to non-migrant workers. In the garment industry, male workers (AOR = 1.95; p = 0.002) and those working over 48 hours per week (AOR = 1.70; p = 0.063) were at greater risk of OI. A hazardous work environment significantly increased the odds of OI in both industries (textile: AOR = 13.06; p < 0.001; and garment: AOR = 3.13; p < 0.001). Additionally, garment workers without adequate protective equipment or cloth while working had a higher likelihood of OI (AOR = 1.90; p = 0.006). Regionally, workers in the Barisal division had higher odds of OI in the textile industry. Although the overall prevalence

**Data availability statement:** All relevant data are within the paper and its Supporting Information files.

**Funding:** The author(s) received no specific funding for this work.

**Competing interests:** The authors have declared that no competing interests exist.

of OI was low, the disproportionate burden among certain subgroups, especially in jute manufacturing and the manufacture of spooling and thread, highlights critical areas for intervention. Improving workplace safety through the provision of protective equipment and a safer working environment is essential to mitigating OI in the textile and garment industries of Bangladesh.

## Background

Occupational injuries (OIs) contribute significantly to both morbidity and mortality. It is estimated that 56 million deaths are attributable to OI, which are estimated to be 5 million (9% of total) global premature deaths every year [1]. According to the International Labor Organization (ILO), due to occupational disease, about 2.78 million workers die every year [2]. Due to work-related hazards, 1.53 million people worldwide lost their lives in 2016 [3]. Additionally, workplace-related injuries resulted in 76.1 million cases of illness globally [3]. The global economic burden due to OIs is 3.94% of the Gross Domestic Product (GDP) per year across the world [4].

In Asia and the Pacific, over 1.2 million deaths occur each year due to workplace-related issues, including inadequate protection, lack of proper uniforms, and insufficient training for workers [5]. Women, younger people, the disabled, overseas workers, tribal communities, untrained, and the poorest workers are at a higher risk and tend to suffer the most in the case of OIs [5,6]. OIs not only affect the individual level but also affect the output of the organization and the society [5]. In Bangladesh, around twenty-five thousand workers pass away due to work environment-related problems, and eight million suffer from injuries from workplace incidents, among them, most of them become permanently disabled [7].

A variety of reasons can lead to occupational injuries [8–15]. Research has indicated that several factors, including physically demanding jobs, long hours and shift work, stress at work, lack of safety training, lack of a safety sign, sleep problems, heavy workload, alcohol consumption, cigarette smoking, poor exercise habits, and chewing khat, increase the risk of OIs [8–14]. Job tenure has an effect on workplace injuries as well [15]. Researchers discovered differences in injury risks based on age [15–17], sex [16], education [18], race/ethnicity [19], and geographical area [20]. Younger workers usually have little experience, which may raise their risk of OIs [15]. Falling (33%), being struck by an object (11%), electrocution (9%), and becoming trapped in or between objects (6%) are the most frequent occupational fatal injuries occurring during construction tasks [21]. Furthermore, initiatives aimed at preventing work-related illnesses and injuries hardly ever consider the opinions and experiences of workers regarding occupational health and safety [22].

Occupational fatal accidents are a worrying problem in Bangladesh, especially in the ready-made garment (RMG) industry. Between 2005 and 2013, at least 1,512 workers lost their lives in various industrial mishaps in RMG industry alone, including building collapse, fire, and riots [23]. In addition, around 10,259 garment workers suffered injuries in accidents [23]. Statistics show that thousands of people are killed

and injured annually in work-related incidents [24]. These mishaps result in enormous financial losses [25] as well as unimaginable pain for the employers, employees, and their families [26]. Along with financial losses, death or injury also has a detrimental effect on businesses due to increased staff turnover, absenteeism, insurance costs, workers' compensation, and decreased productivity [27].

In Bangladesh, the garment sector stands out as the highest earner of remittances and employs the largest workforce. Likewise, the textile industry play a significant role in contributing to the country's GDP, employing a substantial number of people. To our knowledge, evidence on OIs in Bangladesh is scarce. Also, there is a lack of information about the associated factors of OIs in the garment and textile industries. Therefore, in this study, we aimed to investigate the prevalence and associated factors of OI. Identification of associated factors of OI among workers in the garment and textile industries will help garment owners to take precautionary measures in the workplace to ensure safety. Also, it will assist policymakers in developing policies to protect the rights, safety, and security of workers in the garment and textile industries and assist law enforcement bodies in identifying and addressing the hazards linked to OI.

## Materials and methods

### Data source

The data were extracted from the Bangladesh Labor Force Survey (BLFS) 2016–2017. The LFS data is not publicly accessible. However, we acquired the data from the Bangladesh Bureau of Statistics (BBS) following their data-sharing policy and associated cost. This cross-sectional survey has been conducted by the BBS, the National Statistical Organization of the country, with the support of the World Bank. We included data from all quarterly surveys of the years 2016–2017, which included 13,738 participants. Among them, 3,793 were from the textile industry and 9,945 were from the garment industry. The BLFS collected a variety of information, including demographic, labor force, non-economic activities, etc.

### Sampling design

The BLFS used the sampling frame based on the Population and Housing Survey 2011. The sampling frame consists of Enumeration Areas, which are used for preparing the primary sampling unit (PSU). On average, each PSU consisted of 225 households. The BLFS applied two-stage sampling techniques for selecting the respondents. In the first stage, the primary sampling units were selected randomly. In the second stage, a systematic random sampling procedure was applied to select 24 households from each PSU. The survey included about 1,23,000 households all over the country across seven administrative divisions. The detailed sampling design and methodologies can be found in the survey report [28].

### Outcome measure

The outcome variable of this study was occupational injury. Occupational injury is defined as any personal injury (e.g., bruises, minor cuts, burns, amputations, and fatalities) resulting from a work-related accident experienced in the last 12 months before the survey. If the respondents reported any injuries, we categorized it as "yes"; otherwise, "no". The coding for analysis was 1 for "yes" and 0 for "no".

### Independent variables

In this study, a set of covariates including respondent's age (15–24 years, 25–44 years, 45–54 years and ≥55 years), sex of the respondents (Male, Female), respondent's current marital status (Unmarried, Married and Widow/Widower/Separated/Divorced), respondent's education (No formal education, primary, secondary or higher), religion (Muslim and other including Hindu, Buddhist, and Christian), migration status (non-migrant, migrant), job status (formal workers, informal workers), job type (part-time, full time), total working hour in a week (≤ 48 hours, > 48 hours), worked any day at night in last week (yes, no), protected by equipment or cloth during working (yes, no), extreme cold or heating last 12 months

(yes, no), use dangerous tools knives, blades (yes, no), work underground or at heights (yes, no), workplace too dark or confined/insufficient ventilation (yes, no), work in hazardous environment (yes, no), sexually abused or touched (yes, no), constantly shouted at/ repeatedly insulted (yes, no), beaten/physically hurt (yes, no), abused at work place (yes, no), wealth quintiles (Poorest, Poorer, Middle, Richer, and Richest), and respondent's current employment status (Unemployed, Employed). We also considered administrative divisions (Barisal, Chittagong, Dhaka, Khulna, Rajshahi, Rangpur, and Sylhet) and type of place of residence (Urban, Rural) as potential covariates.

## Statistical analysis

Univariate analysis was performed and presented the estimates in frequencies with percentages. A bar chart was used to visualize the different categories of OIs among respondents in the textile and garment industries. Bivariate analysis using a simple Firth logistic regression model was employed to measure the association of OI with individual and job-related characteristics. We used Firth logistic regression due to the methodological advantage of low prevalence of outcome over the traditional logistic regression [29]. The results of the Firth logistic regression were displayed as unadjusted odds ratios with 95% confidence intervals (CIs). Finally, multiple Firth binary logistic regression analysis was carried out to explore the associated factors of OI. The strength of the association of multiple Firth logistic regression was measured in adjusted odds ratios (AORs) with 95% CIs. In the multiple regression model, we entered the variables that were found to be significant (p-value <0.05) in simple regression models. We checked the multicollinearity among the independent variables, and we excluded the correlated variables from the model. Stata software version 15.0 was used to analyze the data.

## Ethical statements

The BBS carried out the BLFS following the 2013 Statistics Act. Planning, carrying out, evaluating, and producing the report were all mostly done by the BBS. The BBS was the implementing agency and was in charge of fieldwork and data processing. The BBS and the Statistics and Informatics Division (SID) oversaw day-to-day technical operations, including hiring, training, and managing field and office workers as well as data processing and field personnel. The World Bank made a major contribution to the development of the questionnaire and sample design for the initial Rotational Panel Sample Design. The BBS officers performed in-person interviews as well as hired enumerators who underwent specialized training for the survey to collect data. Using structured questionnaires, they visited the selected households to collect data on labor force participation, non-economic activities, and demographics. Before conducting the interview, verbal consent was taken from the respondents. The information collected from the respondents was deidentified. Field checks were conducted by the experienced BBS and SID officers to verify any erroneous information gathered from interviewees. Data quality was assured by reinterviews among the selected households. Inclusive training programs were held for survey coordinators, enumerators, and supervisors to ensure the survey's effectiveness [28].

## Results

### Sample characteristics

Among the study participants in the textile industry, 22.5% and 15% belonged to the 15–24- and 45–54-year age groups, respectively. On the other hand, in the garment industry, 36.7% and only 5% of participants belonged to these age groups, respectively. About 40% of the respondents in the textile industry were female, whereas it was about 47% in the garment industry. In the textile industry, 30.5% of respondents were illiterate, whereas it was only 12% in the garment industry. Only 28% of participants in textile industry migrated from different areas, but in garments, it was 63.2%. About 41% of participants in textile industry worked less than or equal to 48 hours last week; on the other hand, it was about 22% in the garments. In the case of the wealth index, in textile industry, 15.9% and 24.2% of participants belonged to the poorest and poorer groups, respectively. On the other hand, in the garments, the percentages were somewhat low (8.3% in the

poorest and 8.2% in the poorer group). In textiles, 33.4%, 18.1%, and 31.8% of respondents were from Dhaka, Khulna, and Rajshahi divisions, respectively, whereas in garments, notably 69.7% and 17.2% of respondents were from Dhaka and Chittagong divisions (**Table 1**).

## Prevalence of occupational injury

Overall, the prevalence of occupational injury among the study participants was 1.8%, with 3.8% in the textile industry and 1.1% in the garment industry. Within the textile industry, the highest prevalence was the manufacture of jute industries (12.3%), followed by the manufacture of spooling and thread industries (7.4%), the manufacture of cordage, rope, twine, and netting industries (3.6%), and pressing and belling jute industries (3.6%). On the other hand, within the garment industry, the highest prevalence was embroidery of textile goods and wearing industries (1.8%), and the lowest prevalence was the manufacture of knitted and crocheted apparel industries (0.7%) (**Fig 1**).

## Bivariate associated factors of occupational injury

**Individual factors.** Respondent's age (45–54 years and ≥55 years; $p < 0.01$) was associated with OI among the participants in the textile industry. Prevalence of OI was higher among the richer (5.7%) and middle group (5.3%) compared to the richest group of the participants in the textile industry (OR= 3.0, $p = 0.001$, 95% CI = 1.6, 5.6 for richer and OR=2.8, $p = 0.001$, 95% CI = 1.5, 5.3 for the middle group). The prevalence of OI was also higher among migrant workers (6.3%) compared to non-migrant workers (0.8%) of the study participants in the textile industry (OR=2.6, $p < 0.001$, 95% CI = 1.8, 3.6). Among the study participants in the textile industry, the likelihood of OI was higher in Barisal, Khulna and Rangpur division compared to the Dhaka division (OR=39.0, $p < 0.001$, 95% CI = 18.0, 84.9 for Barisal, OR=16.1, $p < 0.001$, 95% CI = 8.6, 30.0 for Khulna, and OR=4.5, $p < 0.001$, 95% CI = 1.9, 10.5 for Rangpur division). In the case of participants in the garment industry, we found a higher likelihood of OI among participants living in the Rangpur division compared to the Dhaka division (OR=4.2, $p < 0.001$, 95% CI = 2.6, 6.8) (**Table 2**).

**Job-related factors.** In the textile industry, the prevalence of OI was less (2.9%) among workers who worked more than 48 hours in the last week compared to workers who worked less than or equal to 48 hours (4.7%) (OR=0.6, $p = 0.004$, 95% CI = 0.4, 0.9). In contrast, in the garment industry, the prevalence was higher among workers who worked more than 48 hours (1.2%) in the last week compared to the workers who worked less than or equal to 48 hours (0.7%) in the last week (OR=1.8, $p = 0.038$, 95% CI = 1.0, 3.0). The likelihood of OI was higher among the participants of both textile and garment industries who were protected by equipment or cloth during working (OR=4.0, $p = 0.005$, 95% CI = 1.5, 10.1 for textile and OR=1.6, $p = 0.034$, 95% CI = 1.0, 2.4 for garment industries). We also found a higher prevalence of OI among the participants of both textile (10.9%) and garment (2.2%) industries who worked in a hazardous environment (OR=14.9, $p < 0.001$, 95% CI = 9.4, 23.6 for textile and OR=2.7, $p < 0.001$, 95% CI = 1.9, 4.0 for garment industries) (**Table 2**).

## Multivariable associated factors

In multivariable Firth logistic regression models, we included covariates that were statistically significant ($p < 0.05$) in the simple Firth logistic regression model, along with some important non-significant variables, such as, respondent's sex. Although statistically significant in the simple model, we excluded place of residence and marital status from the multivariable model, as they were highly correlated with the migration status and age of the respondents, respectively. Adjusted models revealed that, in the textile industry, the likelihood of OI was 65% higher among migrant workers (AOR = 1.65, $p = 0.017$, 95% CI = 1.09, 2.50) compared to non-migrant workers. The likelihood of OI was 2.57 times higher among textile workers (AOR = 2.57, $p = 0.087$, 95% CI = 0.87, 5.58) and 1.90 times higher among garment workers (1.90, $p = 0.006$, 95% CI = 1.20, 3.00) who used protective equipment or clothing while working. The probability of OI was 13 times higher among textile workers (AOR = 13.06, $p < 0.001$, 95% CI = 7.84, 21.76) and 3 times higher among garment workers (3.13, $p < 0.001$, 95% CI = 2.08, 4.71) who worked in a hazardous

**Table 1. Background characteristics of the study participants.**

| Characteristics | Textiles (N = 3793) | Garments (N = 9945) |
|---|---|---|
|  | n (%) | n (%) |
| **Age of respondents (in years)** | | |
| 15–24 | 852 (22.5) | 3652 (36.7) |
| 25–44 | 2037 (53.7) | 5641 (56.7) |
| 45–54 | 567 (15.0) | 496 (5.0) |
| ≥ 55 | 337 (8.9) | 156 (1.6) |
| **Sex of respondents** | | |
| Female | 1546 (40.8) | 4767 (47.9) |
| Male | 2247 (59.2) | 5178 (52.1) |
| **Marital status** | | |
| Unmarried | 651 (17.1) | 2436 (24.5) |
| Married | 2958 (78.0) | 7059 (71.0) |
| Widow/widower/separated/divorced | 184 (4.9) | 450 (4.5) |
| **Education level of respondents** | | |
| No formal education | 1158 (30.5) | 1195 (12.0) |
| Primary | 1068 (28.2) | 3140 (31.6) |
| Secondary or higher | 1567 (41.3) | 5610 (56.4) |
| **Migration status** | | |
| Migrant | 1063 (28.0) | 6281 (63.2) |
| Non-Migrant | 2730 (72.0) | 3664 (36.8) |
| **Total working hours in last week** | | |
| ≤ 48 hours | 1565 (41.3) | 2225 (22.4) |
| > 48 hours | 2228 (58.7) | 7720 (77.6) |
| **Wealth index** | | |
| Poorest | 604 (15.9) | 828 (8.3) |
| Poorer | 917 (24.2) | 813 (8.2) |
| Middle | 827 (21.8) | 1785 (18.0) |
| Richer | 814 (21.5) | 3021 (30.4) |
| Richest | 631 (16.6) | 3498 (35.1) |
| **Administrative division** | | |
| Dhaka | 1265 (33.4) | 6932 (69.7) |
| Barisal | 73 (1.9) | 134 (1.4) |
| Chittagong | 270 (7.1) | 1712 (17.2) |
| Khulna | 687 (18.1) | 164 (1.7) |
| Rajshahi | 1206 (31.8) | 359 (3.6) |
| Rangpur | 263 (6.9) | 587 (5.9) |
| Sylhet | 29 (0.8) | 57 (0.5) |
| **Types of the place of residence** | | |
| Rural | 2026 (53.4) | 2673 (26.9) |
| Urban | 1767 (46.6) | 7272 (73.1) |

environments. Moreover, among workers in the textile industry, the odds of OI were higher for those who living in Barisal (AOR = 30.44, p < 0.001, 95% CI = 12.09, 76.68), Khulna (AOR = 6.09, p < 0.001, 95% CI = 2.96, 12.51), and Rangpur (AOR = 5.33, p < 0.001, 95% CI = 2.10, 13.53) divisions, but lower in Rajshahi (AOR = 0.14, p = 0.008, 95%

**Fig 1.  Prevalence of occupational injury across different types of textiles and garments industries.**

CI = 0.03, 0.60) division, compared to those living in Dhaka division. On the other hand, the odds of OI was 9 times higher among the participants of garment industry living in Rangpur division compared to those in Dhaka division (AOR = 9.44, p < 0.001, 95% CI = 4.45, 20.03). Furthermore, among the participants in the garment industry, male participants and those who worked more than 48 hours had 95% (AOR = 1.95, p = 0.002, 95% CI = 1.27, 2.99) and 70% (AOR = 1.70, p = 0.063, 95% CI = 0.97, 2.99) higher odds of OI, respectively compared to female participants and those who worked ≤48 hours in the last week (**Table 3**).

## Discussion

To our knowledge, this is the first study in Bangladesh to examine the prevalence and associated factors of OI among textile and garment workers using nationally representative survey data. In this study, the overall OI prevalence was less than two percent, higher in the textile industry than in the garment industry. The result indicates a lower prevalence of OI compared to research findings from Ethiopia (31.4% and 42.7%) [30,31] and Turkey (65.8%) [32]. The number of participants in each study, as well as the infrastructure and safety culture involved, could be the cause of the variation in the results of OI. Following the research findings, previous studies have revealed that the textile industry is the most prevalent

**Table 2. Distribution and associated factors of occupational injury using simple firth logistic regression models.**

| Variables | Textiles (n = 3793) | | | | Garments (n = 9945) | | | |
|---|---|---|---|---|---|---|---|---|
| | N | *% | COR (95% CI) | p-value | N | **% | COR (95% CI) | p-value |
| **Age of respondents (in years)** | | | | | | | | |
| 15–24 | 852 | 2.5 | Ref. | | 3652 | 1.1 | Ref. | |
| 25–44 | 2037 | 3.3 | 1.4 (0.8, 2.2) | 0.263 | 5641 | 1.2 | 1.1 (0.8, 1.7) | 0.560 |
| 45–54 | 567 | 5.3 | 2.2 (1.3, 3.9) | 0.006 | 496 | 0.4 | 0.5 (0.1, 1.7) | 0.238 |
| ≥ 55 | 337 | 5.6 | 2.4 (1.3, 4.5) | 0.007 | 156 | 0.0 | 0.3 (0.1, 4.8) | 0.388 |
| **Sex of respondents** | | | | | | | | |
| Female | 1546 | 3.1 | Ref. | | 4767 | 0.8 | Ref. | |
| Male | 2247 | 4.0 | 1.3 (0.9, 1.8) | 0.172 | 5178 | 1.4 | 1.8 (1.2, 2.7) | 0.004 |
| **Marital Status** | | | | | | | | |
| Unmarried | 651 | 1.5 | Ref. | | 2436 | 1.2 | Ref. | |
| Married | 2958 | 4.0 | 2.7 (1.4, 4.9) | 0.004 | 7059 | 1.1 | 0.9 (0.7, 1.3) | 0.499 |
| Widow/widower/ separated/divorced | 184 | 4.4 | 2.9 (1.1, 7.4) | 0.021 | 450 | 0.7 | 0.6 (0.2, 1.9) | 0.394 |
| **Education** | | | | | | | | |
| No formal education | 1158 | 3.5 | Ref. | | 1195 | 0.7 | Ref. | |
| Primary | 1068 | 3.3 | 0.9 (0.6, 1.5) | 0.737 | 3140 | 1.1 | 1.6 (0.7, 3.3) | 0.253 |
| Secondary or higher | 1506 | 3.9 | 1.1 (0.7, 1.6) | 0.644 | 5610 | 1.2 | 1.7 (0.8, 3.5) | 0.146 |
| **Migration status** | | | | | | | | |
| Non-Migrant | 2730 | 2.6 | Ref. | | 3664 | 0.9 | Ref. | |
| Migrant | 1063 | 6.3 | 2.6 (1.8, 3.6) | <0.001 | 6281 | 1.2 | 1.3 (0.9, 1.9) | 0.232 |
| **Job-status** | | | | | | | | |
| Formal workers | 553 | 2.7 | Ref. | | 369 | 1.4 | Ref. | |
| Informal workers | 3240 | 3.8 | 1.4 (0.8, 2.3) | 0.256 | 9576 | 1.1 | 0.7 (0.3, 1.7) | 0.477 |
| **Job type** | | | | | | | | |
| Part-time | 844 | 0.7 | Ref. | | 733 | 1.4 | Ref. | |
| Full- time | 2949 | 4.4 | 6.0 (2.7, 13.3) | <0.001 | 9212 | 1.1 | 0.8 (0.4, 1.4) | 0.384 |
| **Total working hours in last week** | | | | | | | | |
| ≤ 48 hours | 1565 | 4.7 | Ref. | | 2225 | 0.7 | Ref. | |
| > 48 hours | 2228 | 2.9 | 0.6 (0.4, 0.9) | 0.004 | 7720 | 1.2 | 1.8 (1.0, 3.0) | 0.038 |
| **Worked any day at night in the last week** | | | | | | | | |
| No | 3618 | 3.6 | Ref. | | 9559 | 1.1 | Ref. | |
| Yes | 175 | 4.6 | 1.4 (0.7, 2.8) | 0.388 | 386 | 1.3 | 1.3 (0.6, 3.1) | 0.546 |
| **Protected by equipment or cloth during working** | | | | | | | | |
| Yes | 430 | 0.9 | Ref. | | 3829 | 0.8 | Ref. | |
| No | 3363 | 4.0 | 4.0 (1.5, 10.1) | 0.005 | 6116 | 1.3 | 1.6 (1.0, 2.4) | 0.034 |
| **Extreme cold or heating last 12 months** | | | | | | | | |
| No | 3639 | 2.8 | Ref. | | 9739 | 1.0 | Ref. | |
| Yes | 154 | 22.1 | 9.8 (6.4, 15.0) | <0.001 | 206 | 3.9 | 4.1 (2.0, 8.3) | <0.001 |
| **Dangerous tools knives, blades** | | | | | | | | |
| No | 3135 | 1.5 | Ref. | | 8967 | 0.8 | Ref. | |
| Yes | 658 | 13.7 | 10.4 (7.2, 14.9) | <0.001 | 978 | 3.6 | 4.5 (3.0, 6.7) | <0.001 |
| **Work underground or at heights** | | | | | | | | |
| No | 3766 | 3.5 | Ref. | | 9932 | 1.1 | Ref. | |
| Yes | 27 | 14.8 | 5.2 (1.9, 14.5) | 0.002 | 13 | 7.7 | 10.9 (2.0, 59.7) | 0.006 |

*(Continued)*

**Table 2.** (Continued)

| Variables | Textiles (n=3793) | | | | Garments (n=9945) | | | |
|---|---|---|---|---|---|---|---|---|
| | N | *% | COR (95% CI) | p-value | N | **% | COR (95% CI) | p-value |
| **Workplace too dark or confined/insufficient ventilation** | | | | | | | | |
| No | 3643 | 2.4 | Ref. | | 9808 | 1.0 | Ref. | |
| Yes | 150 | 32.0 | 18.8 (12.6, 28.1) | <0.001 | 137 | 8.8 | 9.9 (5.4, 18.4) | <0.001 |
| **Work in hazardous environment** | | | | | | | | |
| No | 2742 | 0.8 | Ref. | | 7945 | 0.8 | Ref. | |
| Yes | 1051 | 10.9 | 14.9 (9.4,23.6) | <0.001 | 2000 | 2.2 | 2.7 (1.9, 4.0) | <0.001 |
| **Sexually abused (touched)** | | | | | | | | |
| No | 3767 | 3.6 | Ref. | | 9823 | 1.1 | Ref. | |
| Yes | 26 | 7.7 | 2.7 (0.7, 10.2) | 0.133 | 122 | 0.8 | 1.1 (0.2, 5.6) | 0.904 |
| **Constantly shouted at/ repeatedly insulted** | | | | | | | | |
| No | 3674 | 3.6 | Ref. | | 8957 | 1.1 | Ref. | |
| Yes | 119 | 5.0 | 1.5 (0.7, 3.5) | 0.293 | 988 | 0.7 | 0.7 (0.3, 1.4) | 0.274 |
| **Beaten/physically hurt** | | | | | | | | |
| No | 3788 | 3.6 | Ref. | | 9911 | 1.1 | Ref. | |
| Yes | 5 | 0.0 | 2.4 (0.1, 43.9) | 0.551 | 34 | 0.0 | 1.1 (0.2, 5.6) | 0.904 |
| **Abused at workplace** | | | | | | | | |
| No | 3644 | 3.5 | Ref. | | 8816 | 1.2 | Ref. | |
| Yes | 149 | 5.4 | 1.6 (0.8, 3.3) | 0.179 | 1129 | 0.7 | 0.7 (0.3, 1.3) | 0.231 |
| **Wealth index** | | | | | | | | |
| Richest | 631 | 1.9 | Ref. | | 3498 | 1.1 | Ref. | |
| Richer | 814 | 5.7 | 3.0 (1.6, 5.6) | 0.001 | 3021 | 1.0 | 0.9 (0.6, 1.4) | 0.640 |
| Middle | 827 | 5.3 | 2.8 (1.5, 5.3) | 0.001 | 1785 | 1.1 | 0.9 (0.6, 1.7) | 0.903 |
| Poorer | 917 | 2.5 | 1.3 (0.7, 2.6) | 0.456 | 813 | 1.0 | 0.9 (0.4, 1.9) | 0.836 |
| Poorest | 604 | 2.0 | 1.1 (0.5, 2.3) | 0.912 | 828 | 1.6 | 1.5 (0.8, 2.7) | 0.242 |
| **Administrative division** | | | | | | | | |
| Dhaka | 1265 | 0.9 | Ref. | | 6932 | 1.0 | Ref. | |
| Barisal | 73 | 26.0 | 39.0 (18.0, 84.9) | <0.001 | 134 | 0.0 | 0.4 (0.02, 6.1) | 0.494 |
| Chittagong | 270 | 2.2 | 2.7 (1.0, 7.1) | 0.047 | 1712 | 0.9 | 0.9 (0.6, 1.7) | 0.969 |
| Khulna | 687 | 12.8 | 16.1 (8.6, 30.0) | <0.001 | 164 | 0.6 | 0.9 (0.2, 4.7) | 0.933 |
| Rajshahi | 1206 | 0.2 | 0.2 (0.1, 0.9) | 0.034 | 359 | 0.6 | 0.7 (0.2, 2.5) | 0.598 |
| Rangpur | 263 | 3.8 | 4.5 (1.9, 10.5) | <0.001 | 587 | 3.9 | 4.2 (2.6, 6.8) | <0.001 |
| Sylhet | 29 | 3.5 | 5.7 (1.1, 32.8) | 0.049 | 57 | 0.0 | 0.9 (0.1, 14.5) | 0.931 |
| **Types of the place of residence** | | | | | | | | |
| Rural | 2026 | 2.4 | Ref. | | 2673 | 1.1 | Ref. | |
| Urban | 1767 | 5.0 | 2.2 (1.5, 3.1) | <0.001 | 7272 | 1.1 | 1.1 (0.7, 1.6) | 0.817 |

*Prevalence of occupational injury among the workers of the textile industry; **Prevalence of occupational injury among the workers of the garment industry; COR: Crude Odds Ratio; CI: Confidence Interval

manufacturing enterprise with a high incidence of work-related injuries [33,34]. These injuries can range from manual handling and operating potentially hazardous machinery to exposure to noise and hazardous substances [33,35,36]. Whereas, equal hazards and accidents were found in the textile and RMG sectors, with the former study conducted in Bangladesh reporting rates of OI less than 5% [37].

**Table 3. Associated factors of occupational injury using multiple firth logistic regression models.**

| Variables | Textiles | | Garments | |
|---|---|---|---|---|
| | AOR (95% CI) | p-value | AOR (95% CI) | p-value |
| **Age of respondents (in years)** | | | | |
| 15–24 | Ref. | | Ref. | |
| 25–44 | 1.15 (0.65, 2.02) | 0.641 | 1.13 (0.75, 1.69) | 0.561 |
| 45–54 | 1.19 (0.61, 2.30) | 0.612 | 0.44 (0.12, 1.61) | 0.215 |
| ≥ 55 | 1.28 (0.60, 2.71) | 0.520 | 0.32 (0.02, 5.22) | 0.422 |
| **Sex of respondents** | | | | |
| Female | Ref. | | Ref. | |
| Male | 0.83 (0.53, 1.30) | 0.418 | 1.95 (1.27, 2.99) | 0.002 |
| **Migration status** | | | | |
| Non-Migrant | Ref. | | Ref. | |
| Migrant | 1.65 (1.09, 2.50) | 0.017 | 2.98 (1.60, 5.53) | 0.001 |
| **Total Working hours in last week** | | | | |
| ≤ 48 hours | Ref. | | Ref. | |
| > 48 hours | 0.79 (0.53, 1.18) | 0.253 | 1.70 (0.97, 2.99) | 0.063 |
| **Protected by equipment or cloth during working** | | | | |
| Yes | Ref. | | Ref. | |
| No | 2.57 (0.87, 7.58) | 0.087 | 1.90 (1.20, 3.00) | 0.006 |
| **Work in hazardous environment** | | | | |
| No | Ref. | | Ref. | |
| Yes | 13.06 (7.84, 21.76) | <0.001 | 3.13 (2.08, 4.71) | <0.001 |
| **Wealth Index** | | | | |
| Richest | Ref. | | Ref. | |
| Richer | 1.05 (0.52, 2.12) | 0.891 | 0.78 (0.48, 1.27) | 0.312 |
| Middle | 1.28 (0.63, 2.60) | 0.503 | 0.84 (0.46, 1.51) | 0.553 |
| Poorer | 1.06 (0.47, 2.35) | 0.895 | 0.75 (0.30, 1.88) | 0.544 |
| Poorest | 1.01 (0.39, 2.57) | 0.891 | 0.80 (0.34, 1.87) | 0.600 |
| **Division** | | | | |
| Dhaka | Ref. | | Ref. | |
| Barisal | 30.44 (12.09, 76.68) | <0.001 | 0.72 (0.04, 12.4) | 0.824 |
| Chittagong | 1.59 (0.57, 4.47) | 0.380 | 0.76 (0.42, 1.36) | 0.350 |
| Khulna | 6.09 (2.96, 12.51) | <0.001 | 1.39 (0.26, 7.52) | 0.700 |
| Rajshahi | 0.14 (0.03, 0.60) | 0.008 | 1.29 (0.33, 5.05) | 0.714 |
| Rangpur | 5.33 (2.10, 13.53) | <0.001 | 9.44 (4.45, 20.1) | <0.001 |
| Sylhet | 3.83 (0.59, 24.63) | 0.188 | 1.36 (0.08, 23.9) | 0.833 |
| Average variance inflation factor | 1.14 | | 1.14 | |
| Area under the receiver operating characteristic curve | 0.918 | | 0.659 | |

AOR: Adjusted Odds Ratio; CI: Confidence Interval.

The highest prevalence of OI was in the manufacture of jute industries (12.3%), and the lowest prevalence was found in the manufacture of knitted and crocheted apparel industries (0.7%). In line with the results of this study, Sah D. P. & Mishra A. K. [38] concluded that the rate of accidents in jute mills is high. The most prevalent injury was squeezing off a single joint or the entire finger. A hand or arm being lost entirely or in part while operating machinery was also quite prevalent. The majority of jute mill workers reported experiencing accidents at least once during their careers. In contrast, the

results of a previous study showed that the health hazards were considerably less for jute workers than for garment sector workers [39].

In the textile industry, migration status, working environment, and administrative division were significantly associated with OI. Comparably, an Indian study found that migrant workers in the textile industry suffered disproportionately [40]. Although there is limited published research on the matter, the majority of data indicate that migrant workers face both a high occupational risk and a high frequency of fatalities [41,42]. Research indicated that workers in the textile industry are susceptible to a variety of occupational health hazards, with the hands being the most frequently injured body part [30,31]. This might happen considering these bodily parts are the most active and regularly come into contact with different tools and machinery [43].

In the garment industry, on the other hand, the sex of the respondent, total working hours, protection during working, working environment, and administrative division were significantly associated with OI. A study in Ethiopia found that gender was significantly associated with non-fatal OI [34], which is similar to our finding. We observed that male workers were more likely than females to incur an OI, which is consistent with previous research among Ethiopian textile industry workers [6,31,34,44]. It was also discovered through studies conducted in Tanzania and Mexico that male workers had higher physical injury than female workers [45,46]. The explanation could be that men are more likely than women to engage in risk-taking activities, such as taking on risky tasks [47,48]. Another reason could be that female workers are typically assigned to less dangerous departments like knitting and clothing rather than weaving and dyeing [31]. Conversely, a study indicates that compared to male workers, female workers were substantially more likely to experience occupational stress and health risks [49]. Concerning the overall number of working hours, we observed that the number of hours worked per week was highly correlated with OI. This finding is consistent with research from Ethiopia, Japan, and Thailand that revealed the chance of injury was significantly impacted by the number of hours worked per week [33,50–52]. Workers who worked for more than 48 hours had a 2.3-fold increased risk of a work-related injury compared to those who worked for the same or less than 48 hours [53]. The worker may have eye strain if he concentrates for an extended amount of time, which could lead to a loss of focus and subsequent harm [33]. In contrast, research found no significant variation in injury rates based on hours worked per week, which could be given to the limited number of workers who work more than 48 hours per week [6,30].

Other factors, such as the non-availability of personal protective equipment (PPE), the lack of suitability of PPE, and employers failing to provide PPE, have been shown to have an effect on health concerns [34,54–56], which is consistent with our study findings. Some issues that arise at work appear to be related to the surroundings in which workers operate. Similar to our study finding, Martinelli K. [57] reported that environmental factors can cause OI. Operators suffer vision problems due to the low lighting at their workstations and a lack of eye-protective glasses [57]. Jahan M. [58] discovered similar environmental problems in her research of five garment manufacturers. In their study, Joshi et al. [59] demonstrated that long working hours, unsafe working conditions, a lack of supervision and training, the use of old machinery and equipment, and a packed manufacturing facility in a highly congested space were significantly associated with occupational hazards. Regular workplace supervision and health and safety training programs may help improve the present condition of the workplace and its workers.

## Limitations of the study

This study has several limitations. First, it included only the garment and textile industries, excluding other sectors that may have different patterns of OI. Second, we used a subsample from the total sample of LFS, which may limit the representativeness of the entire country. Finally, due to the cross-sectional nature of the study, we were unable to establish the cause-effect relationships between OI and independent factor. However, cross-sectional studies are commonly used to explore the association between outcome and independent factor.

**Implications and future directions of this study**

Though our findings reveal a low level of OI among workers of garment and textile industries in Bangladesh, ignoring this could be a huge burden in future. By addressing the factors associated with OI, such as hazardous working conditions and less protective equipment while working, might have an opportunity to create safer and healthier workplaces. These results could be a wake-up call for employers, policymakers, and industry stakeholders to prioritize safety for the workers. Ensuring proper safety measures, arranging periodic training related to safety measures, and enforcing labor laws could be effective ways to lower OI.

While this study provides important findings, there is still much to explore. Future research should include workers from other industries to capture a more complete picture of OI risks across the country. Studies that follow workers over time would help uncover not just what's happening, but why. Understanding these patterns more deeply can lead to better-targeted interventions that protect workers from any form of OIs.

## Conclusion

In Bangladesh, the prevalence of OI is low, about 2%, but it is higher in the textile industry than in the garment industry. Within the textile industry, the highest prevalence was the manufacture of jute textiles. Whereas within the garment industry, the highest prevalence of OI was in the embroidery of textile goods and the wearing industries. Workers who worked in hazardous conditions and weren't protected by equipment or clothing while working had a higher risk of OI. Ensuring protected equipment during work and improving the working environment would help reduce the OI in the textile and garment industries.

## Supporting information

**S1 Data.  Dataset.**
The labor force dataset used for this study in stata file.
(RAR)

## Author contributions

**Conceptualization:** Md. Tariqujjaman, Md. Mehedi Hasan, Md. Rashidul Azad.

**Data curation:** Md. Tariqujjaman, Abul Hares, Fahmida Ferdous, Md. Mehedi Hasan, Md. Rashidul Azad.

**Formal analysis:** Md. Tariqujjaman, Abul Hares, Md. Mehedi Hasan.

**Investigation:** Alamgir Hossain, Md. Mehedi Hasan, Md. Rashidul Azad.

**Methodology:** Md. Tariqujjaman, Arifa Farzana Tanha, Alamgir Hossain, Abul Hares, Md. Matiur Rahaman, Nadia Sultana, Fahmida Ferdous, Md. Mehedi Hasan, Md. Rashidul Azad.

**Project administration:** Md. Rashidul Azad.

**Software:** Md. Tariqujjaman, Abul Hares, Nadia Sultana, Md. Mehedi Hasan.

**Supervision:** Md. Tariqujjaman, Md. Matiur Rahaman, Md. Mehedi Hasan, Md. Rashidul Azad.

**Validation:** Arifa Farzana Tanha, Alamgir Hossain, Abul Hares, Md. Matiur Rahaman, Nadia Sultana, Fahmida Ferdous, Md. Mehedi Hasan, Md. Rashidul Azad.

**Visualization:** Md. Tariqujjaman, Abul Hares.

**Writing – original draft:** Md. Tariqujjaman, Arifa Farzana Tanha, Alamgir Hossain, Nadia Sultana, Md. Rashidul Azad.

**Writing – review & editing:** Md. Tariqujjaman, Arifa Farzana Tanha, Alamgir Hossain, Abul Hares, Md Matiur Rahaman, Nadia Sultana, Fahmida Ferdous, Md. Mehedi Hasan, Md. Rashidul Azad.

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
