## [Decision Letter · Decision Letter 0]

7 Oct 2024

PONE-D-24-20868Examining the Prevalence and Individual and Job-Related Contributing Factors of Occupational Injuries in Garment and Textile Industries: Insights from Bangladesh Labour Force Survey 2016-17PLOS ONE

Dear Dr.  Tariqujjaman, 

Thank you for submitting your manuscript to PLOS ONE. After careful consideration, we feel that it has merit but does not fully meet PLOS ONE’s publication criteria as it currently stands. Therefore, we invite you to submit a revised version of the manuscript that addresses the points raised during the review process. Please check attachment for additional comments provided by reviewer 2.

We look forward to receiving your revised manuscript.

Kind regards,

Ilias Mahmud, Ph.D.

Academic Editor

PLOS ONE

Journal Requirements:

2. We note that you have indicated that there are restrictions to data sharing for this study. PLOS only allows data to be available upon request if there are legal or ethical restrictions on sharing data publicly. For more information on unacceptable data access restrictions, please see http://journals.plos.org/plosone/s/data-availability#loc-unacceptable-data-access-restrictions. Before we proceed with your manuscript, please address the following prompts: a) If there are ethical or legal restrictions on sharing a de-identified data set, please explain them in detail (e.g., data contain potentially identifying or sensitive patient information, data are owned by a third-party organization, etc.) and who has imposed them (e.g., a Research Ethics Committee or Institutional Review Board, etc.). Please also provide contact information for a data access committee, ethics committee, or other institutional body to which data requests may be sent. b) If there are no restrictions, please upload the minimal anonymized data set necessary to replicate your study findings to a stable, public repository and provide us with the relevant URLs, DOIs, or accession numbers. For a list of recommended repositories, please see https://journals.plos.org/plosone/s/recommended-repositories. You also have the option of uploading the data as Supporting Information files, but we would recommend depositing data directly to a data repository if possible. We will update your Data Availability statement on your behalf to reflect the information you provide.

Reviewers' comments:

Reviewer's Responses to Questions

**Comments to the Author**

1. Is the manuscript technically sound, and do the data support the conclusions?

Reviewer #1: Yes

Reviewer #2: No

2. Has the statistical analysis been performed appropriately and rigorously? 

Reviewer #1: Yes

Reviewer #2: I Don't Know

3. Have the authors made all data underlying the findings in their manuscript fully available?

Reviewer #1: No

Reviewer #2: No

4. Is the manuscript presented in an intelligible fashion and written in standard English?

Reviewer #1: Yes

Reviewer #2: Yes

5. Review Comments to the Author

Reviewer #1: Abstract:

- The authors should re-state the highlighted expression on lines 41 & 42

Background:

- The highlighted sentence (lines 51-54) is not clear. An expression in that sentence (highlighted) is incorrect.

- The sentence covering lines 56-58 is not clear.

Results:

- Table 1 - The percentages on Age, Marital Status, Profession, Wealth Index and Administration Division are either more than 100% or less.

- The authors did not indicate the "Mechanism of Injury", "Body Parts Affected by Injury" and "Severity of Injury". These would have completed the narrative

- Can you confirm the total/overall injury prevalence of 1.8%?

Discussion:

- Your study was on occupational injuries. Why are you discussing the effect of dust and other hazards on health, which are not injuries (278-283; 288-291)?

- Incorrect expressions were used, as indicated on lines 268, 324, 325 & 330

-

Conclusion:

- The highlighted expression should be modified, as indicated in-text.

Reviewer #2: Dear Author, This report is very important, but I am unclear about the source of the study population based on my understanding of the labor force report. Additionally, I would like to know why you used Firth's penalized logistic regression. I encourage you to discuss these points with a statistician for a better understanding of the analysis-

6. PLOS authors have the option to publish the peer review history of their article (what does this mean? ). If published, this will include your full peer review and any attached files.

**Do you want your identity to be public for this peer review?** For information about this choice, including consent withdrawal, please see our Privacy Policy .

Reviewer #1: No

Reviewer #2: **Yes: ** Hailemichael Mulugeta

---

## [Author Response · Author response to Decision Letter 1]

31 Dec 2024

Reviewer #1: Abstract:

- The authors should re-state the highlighted expression on lines 41 & 42

Response: Thank you for your suggestion. We have revised the highlighted expressions which now read as below (pages 2-3, lines 42-46):

“Although the overall prevalence of OIs was low, the disproportionate burden among certain subgroups, especially in jute manufacturing, highlights critical areas for intervention. Improving workplace safety through protective equipment provision and safer working environment is essential to mitigate OIs in textile and garment industries of Bangladesh.”

Background:

- The highlighted sentence (lines 51-54) is not clear. An expression in that sentence (highlighted) is incorrect.

Response: We have revised the sentence for more clarity. Below is the revision (page 3, lines 53-55)—

Due to work-related hazards, 1.53 million people worldwide lost their lives in 2016. Additionally, workplace-related injuries resulted in 76.1 million cases of illness globally.

- The sentence covering lines 56-58 is not clear.

Response: We have revised the sentence for more clarity. Below is the revision (page 3, lines 57-59)—

In Asia and the Pacific, over 1.2 million deaths occur each year due to workplace-related issues inadequate protection, lack of proper uniforms, and insufficient training for workers.

Results:

- Table 1 - The percentages on Age, Marital Status, Profession, Wealth Index, and Administration Division are either more than 100% or less.

Response: Thanks for your careful eye. The issues under discussion were due to the rounding errors that have been fixed in the revised version.

- The authors did not indicate the "Mechanism of Injury", "Body Parts Affected by Injury" and "Severity of Injury". These would have completed the narrative

Response: Thanks for raising an important issue. Unfortunately, we are not in a position to report estimates/narratives on the issues mentioned due to the lack of data.

- Can you confirm the total/overall injury prevalence of 1.8%?

Response: Yes. The overall injury is 1.8%.

Discussion:

- Your study was on occupational injuries. Why are you discussing the effect of dust and other hazards on health, which are not injuries (278-283; 288-291)?

Response: Thanks for your valuable concerns. We agree with you. We removed these in the revised version.

- Incorrect expressions were used, as indicated on lines 268, 324, 325 & 330

Response: We apologize for the typos. We have revised it as requested which reads as below—

The present result indicates a lower prevalence of OI compared to research findings from Ethiopia (31.4% and 42.7%) [34-36] and Turkey (65.8%) [37] (page 16, line: 264).

In their study, Joshi et al. [64] demonstrated that long working hours, unsafe working conditions, a lack of supervision and training, the use of old machinery and equipment, and a packed manufacturing facility in a highly congested space were significantly associated with occupational hazards. (page 18, line: 317-320).

Whereas among the garment industries, the highest prevalence of OI was embroidery of textile goods and wearing industries.

Conclusion:

- The highlighted expression should be modified, as indicated in-text.

Response: As requested, we have revised the conclusion section as below (page 19, lines: 323-329).

“In Bangladesh, the prevalence of OI is low, about 2%, higher in textile industries than in garment industries. Among the textile industries, the highest prevalence was the manufacture of jute textiles. Whereas among the garment industries, the highest prevalence of OI was embroidery of textile goods and wearing industries. Employees who worked in hazardous conditions and weren’t protected by equipment or clothing while working had a higher chance of OIs. Ensuring protected equipment during work and improving the working environment would help reduce the OI in the textile and garment industries..”

Reviewer #2: Dear Author, This report is very important, but I am unclear about the source of the study population based on my understanding of the labor force report. Additionally, I would like to know why you used Firth's penalized logistic regression. I encourage you to discuss these points with a statistician for a better understanding of the analysis-

Review manuscript report

Dear Author, This report is very important, but I am unclear about the source of the study population based on my understanding of the labor force report. Additionally, I would like to know why you used Firth's penalized logistic regression. I encourage you to discuss these points with a statistician for a better understanding of the analysis.

line Issues Recommendation Response

Title The title is not a good write-up Prevalence of Occupational Injuries and associated factors related to Individual and Job in Garment and Textile Industries: Insights from Bangladesh Labour Force Survey 2016-17 Thanks a lot for recommending this title: We included it.

26 Rewrite the objective This study aimed to determine the prevalence of occupational injuries and associated factors related to individual and job among garment and textile employees in Bangladesh Added.

29 Analysis with Firth's penalized logistic regression

Firth's penalized logistic regression is used for rare events with a prevalence of < 1% or small data set. Let you look at the following literature

doi: 10.4103/indianjpsychiatry.indianjpsychiatry_827_23

We agree with you. We have a prevalence of 1.2% among garments industries, which is close to 1%. That’s why we used firth logistic regression.

41-43 The recommendation is not based on the findings We have added specific recommendations as per your suggestions. (pages 2-3, lines 42-46):

98 Un clear data source I tried to see the Bangladesh Bureau of Statistics with technical support from the World Bank First Published – January 2018.So, the report did not show the Garment workers related information and one information about textile worker that is a total 1409 worker.

Dear author, it is difficult to understand your source of population based on the following report bellow.

https://mccibd.org/wp-content/uploads/2021/09/Labour-Force-Survey-2016-17.pdf Dear reviewer, I confirm that the data source is the LFS 2016-17. The data is not an open source data. We purchased the data from the Bangladesh Bureau of statistics. We purchased the data by the data purchase policy of BBS. Regarding the garments and textitle information, we categorized these based on the provided codes in the dataset. We will happy to clarify if you have any further concerns.

117 Occupational injury and illness How did you differentiate from data if both were reported as yes and no? Illness is different from injury Agree with you. We dropped illness.

128 workers/service and sales workers/skilled agricultural, forestry and fish, Craft and related trades workers, How did you relate with textile and garment workers? Actually, this is less relevant and we omitted it.

142 Statistical analysis Refer the above issue of line 29 We agree with you. We have a prevalence of 1.2% among garments industries, which is close to 1%. That’s why we used firth logistic regression.

173 Results I am not sure of the data source and it is difficult to deal with it. Mentioned as previous

199 Bivariate associated factors of occupational injuries This part is not stand-alone as a subtopic. But, you can report it in a similar table in different columns of a multivariable report. Thank you for your concern. Presenting bivariate results with multivariable tables will be clumsy. That’s why we presented the bivariate results in a separate table.

263 Discussion I am not sure of the data source and it is difficult to deal with it. Mentioned as previous

---

## [Decision Letter · Decision Letter 1]

20 May 2025

PONE-D-24-20868R1Prevalence of Occupational Injuries and Associated Factors Related to Individual and Job in Garment and Textile Industries: Insights from Bangladesh Labour Force Survey 2016-17PLOS ONE

Dear Dr. Tariqujjaman,

Thank you for submitting your manuscript to PLOS ONE. After careful consideration, we feel that it has merit but does not fully meet PLOS ONE’s publication criteria as it currently stands. Therefore, we invite you to submit a revised version of the manuscript that addresses the points raised during the review process.

We look forward to receiving your revised manuscript.

Kind regards,

Shahnawaz Anwer, PhD

Academic Editor

PLOS ONE

Additional Editor Comments:

Dear Authors

Thank you for your revised manuscript. Your manuscript was reviewed by external reviewer and the academic editor. While the revised manuscript is much improved, there are still some issues, which need to be addressed.

Comments:

1. More than 50% of citations are 10 Y or older. Please update your citations.

2. The definition of occupational injury is not comprehensive. What do you mean by "personal injury". Please specify some of the occupational injuries.

3. Author should discuss limitations of current study

4. Authors should discuss the study implications and future research directions

Reviewers' comments:

Reviewer's Responses to Questions

**Comments to the Author**

1. If the authors have adequately addressed your comments raised in a previous round of review and you feel that this manuscript is now acceptable for publication, you may indicate that here to bypass the “Comments to the Author” section, enter your conflict of interest statement in the “Confidential to Editor” section, and submit your "Accept" recommendation.

Reviewer #2: (No Response)

2. Is the manuscript technically sound, and do the data support the conclusions?

Reviewer #2: Partly

3. Has the statistical analysis been performed appropriately and rigorously? 

Reviewer #2: Yes

4. Have the authors made all data underlying the findings in their manuscript fully available?

Reviewer #2: Yes

5. Is the manuscript presented in an intelligible fashion and written in standard English?

Reviewer #2: Yes

6. Review Comments to the Author

Reviewer #2: Dear Authors,

1. Your title can be shortened for conciseness. Consider revising it to:

“Occupational Injury Prevalence and Associated Factors among Garment and Textile Workers: Insights from the Bangladesh Labour Force Survey 2016-17”

2. Data Analysis:

Please include details about the quality control measures you implemented during the analysis, such as checking for collinearity and assessing model fit.

Best regards.

Hailemichael M.

7. PLOS authors have the option to publish the peer review history of their article (what does this mean? ). If published, this will include your full peer review and any attached files.

**Do you want your identity to be public for this peer review?** For information about this choice, including consent withdrawal, please see our Privacy Policy .

Reviewer #2: **Yes: ** Hailemichael Mulugeta (BSc., MPH)

Assistant Professor of Environmental and Occupational Health

Addis Ababa University

---

## [Author Response · Author response to Decision Letter 2]

4 Jul 2025

Additional Editor Comments:

Dear Authors

Thank you for your revised manuscript. Your manuscript was reviewed by external reviewer and the academic editor. While the revised manuscript is much improved, there are still some issues, which need to be addressed.

Comments:

1. More than 50% of citations are 10 Y or older. Please update your citations.

Response: Thank you so much for your valuable suggestion. We have updated the references.

2. The definition of occupational injury is not comprehensive. What do you mean by "personal injury". Please specify some of the occupational injuries.

Response: According to your suggestion, we have added a few occupational injuries. We have added below (Page 6; lines 122-126)

Occupational injury is defined as any personal injury (e.g., bruises, minor cuts, burns, amputations, and fatalities) resulting from a work-related accident experienced in the last 12 months before the survey. If the respondents reported any injuries, we categorized it as “yes” otherwise “no”. The coding for analysis was 1 for “yes” and 0 for “no”.

3. Author should discuss limitations of current study

Response: According to your suggestion, we have added the limitation section. (Page 19; lines 325-330)

This study has several limitations. First, it included only the garment and textile industries, excluding other sectors that may have different patterns of OI. Second, we used a subsample from the total sample of LFS, which may limit the representativeness of the entire country. Finally, due to the cross-section nature of the study, we were unable to establish the cause-effect relationships between OI and independent factor. However, cross-sectional studies are commonly used to explore the association between outcome and independent factor.

4. Authors should discuss the study implications and future research directions

Response: Thank you so much. Accordingly, we have added the study implications and future research directions. (Page 19; lines 332-343)

Though, our findings reveal a low level of OI among employees of garment and textile industries in Bangladesh, ignoring this could be a huge burden in future. By addressing the factors associated with OI such as hazardous working conditions, and less protecting equipment while working, might have an opportunity to create safer and healthier workplaces. These results could be a wake-up call for employers, policymakers, and industry stakeholders to prioritize safety for the employees. Ensuring proper safety measures, arranging periodic training related to safety measures and enforcing labor laws could be effective ways to lowing OI.

While this study provides important findings, there is still much to explore. Future research should include employees from other industries to capture a more complete picture of OI risks across the country. Studies that follow employees over time would help uncover not just what’s happening, but why. Understanding these patterns more deeply can lead to better-targeted interventions that protect employees from any forms of OIs.

Reviewers' comments:

Comments to the Author

Reviewer #2: Dear Authors,

1. Your title can be shortened for conciseness. Consider revising it to:

“Occupational Injury Prevalence and Associated Factors among Garment and Textile Workers: Insights from the Bangladesh Labour Force Survey 2016-17”

Response: Thank you for your valuable suggestion and input for revising the title. We kept your concise title. However, we have modified a few. Below is the revised title

“Prevalence and Associated Factors of Occupational Injuries among Garment and Textile Employees: Evidence from the Bangladesh Labour Force Survey 2016-17”

2. Data Analysis:

Please include details about the quality control measures you implemented during the analysis, such as checking for collinearity and assessing model fit.

Response: Thank you for your valuable suggestions. We assessed multicollinearity using the Variance Inflation Factor (VIF) and evaluated model fit using the area under the Receiver Operating Characteristic (ROC) curve. Since the Hosmer-Lemeshow test cannot be performed for multiple Firth logistic regression models, we used the AUC of the ROC curve to assess model performance instead. We have included the results of VIF and AUROC curve at the bottom of Table 3 (Page 15). Also, we wrote in the statistical analysis section (Page 7-8, lines 156-157).

“We checked the multicollinearity among the independent variables and we excluded the correlated variables from the model.”

---

## [Editor Report · Decision Letter 2]

9 Jul 2025

PONE-D-24-20868R2Prevalence and Associated Factors of Occupational Injuries among Garment and Textile Employees: Evidence from the Bangladesh Labour Force Survey 2016-17PLOS ONE

Dear Dr. Tariqujjaman,

Thank you for submitting your manuscript to PLOS ONE. After careful consideration, we feel that it has merit but does not fully meet PLOS ONE’s publication criteria as it currently stands. Therefore, we invite you to submit a revised version of the manuscript that addresses the points raised during the review process.

We look forward to receiving your revised manuscript.

Kind regards,

Shahnawaz Anwer, PhD

Academic Editor

PLOS ONE

Journal Requirements:

Additional Editor Comments:

Thank you for your revised manuscript. Manuscript is much improved. However, there are many typo and grammatical errors throughout the manuscript. I would suggest for a language editing by a native speaker.

Minor comments:

Line 66: "injur" should be revised as "injury"

Line 75: "st ruck" should be revised as "struck"

---

## [Author Response · Author response to Decision Letter 3]

21 Aug 2025

Thank you for your revised manuscript. Manuscript is much improved. However, there are many typo and grammatical errors throughout the manuscript. I would suggest for a language editing by a native speaker.

Response: Thank you very much, Sir, for your valuable comments. We tried but were unable to find a native English speaker for the editing. However, we carefully reviewed the text and corrected the grammatical errors.

Minor comments:

Line 66: "injur" should be revised as "injury"

Response: Corrected.

Line 75: "st ruck" should be revised as "struck"

Response: Corrected.

---

## [Editor Report · Decision Letter 3]

2 Sep 2025

Prevalence and Associated Factors of Occupational Injuries among Garment and Textile Workers: Evidence from the Bangladesh Labour Force Survey 2016-17

PONE-D-24-20868R3

Dear Dr. Tariqujjaman,

We’re pleased to inform you that your manuscript has been judged scientifically suitable for publication and will be formally accepted for publication once it meets all outstanding technical requirements.

Kind regards,

Shahnawaz Anwer, PhD

Academic Editor

PLOS ONE
---

## [Editor Report · Acceptance letter]

PONE-D-24-20868R3

PLOS ONE

Dear Dr. Tariqujjaman,

I'm pleased to inform you that your manuscript has been deemed suitable for publication in PLOS ONE. Congratulations! Your manuscript is now being handed over to our production team.

Kind regards,

on behalf of

Dr. Shahnawaz Anwer

Academic Editor

PLOS ONE